# Simple anthropometric measures to predict visceral adipose tissue area in middle-aged Indonesian men

Sahat Basana Romanti Ezer Matondang[1]☯, Bennadi Adiandrian[1]☯, Komang Shary Karismaputri[1], Cicilia Marcella🄳[2], Joedo Prihartono[3], Dicky Levenus Tahapary🄳[2,4]*

1 Department of Radiology, Dr. Cipto Mangunkusumo National General Hospital, Faculty of Medicine Universitas Indonesia, Central Jakarta, Jakarta, Indonesia, 2 Metabolic, Cardiovascular and Aging Cluster, The Indonesian Medical Education and Research Institute, Faculty of Medicine Universitas Indonesia, Central Jakarta, Jakarta, Indonesia, 3 Department of Community Medicine, Faculty of Medicine, Universitas Indonesia, Central Jakarta, Jakarta, Indonesia, 4 Division of Endocrinology, Department of Internal Medicine, Dr. Cipto Mangunkusumo National General Hospital, Faculty of Medicine Universitas Indonesia, Central Jakarta, Jakarta, Indonesia

☯ These authors contributed equally to this work.
* dicky.tahapary@ui.ac.id

**Data Availability Statement:** All relevant data are within the paper and its Supporting information files.

## Abstract

The diagnosing of central obesity requires ethnic-specific cut-offs of waist circumference (WC) and body mass index (BMI). This study aims to develop formulas to predict visceral adipose tissue (VAT) area based on WC and BMI to determine the cut-off points of central obesity in Indonesia. We conducted a cross-sectional study among 32 middle-aged Indonesian men. VAT area was measured using an abdominal CT scan, whereas WC and BMI were assessed through anthropometric measurements. Linear regression analysis was performed to define the formulas to predict VAT area using WC and BMI. Next, the optimal cut-off values of WC and BMI were determined using ROC curve analysis. Strong positive correlations were found between WC and VAT as well as BMI and VAT (r = 0.78; r = 0.67, p <0.001). The formula to predict VAT area from WC was −182.65 + (3.35 × WC), whereas the formula to predict VAT area from BMI was −57.22 + (6.95 × BMI). These formulas predicted WC of 88.5 cm and BMI of 23.9 kg/m² as the optimal cut-off values for central obesity in middle-aged Indonesian men.

## Introduction

Obesity is increasingly becoming a major health concern worldwide. More than 650 million adults worldwide were obese in 2016, and it is predicted that 1 billion people will be obese in 2030, with the fastest rise occurring in low- and middle-income countries (LMIC) [1, 2]. As one of the LMIC, Indonesia is currently suffering from "double burden of disease" where non-communicable diseases are becoming as common as communicable diseases [3]. A study by Harbuwono in 2018 showed that the prevalence of obesity and central obesity have reached

**Funding:** The publication of this study is funded by Universitas Indonesia 2Q2 Grant No. NKB-3385/UN2.RST/HKP.05.00/2020. The funder had no role in study design, data collection and analysis, decision to publish, or preparation of the manuscript.

**Competing interests:** The authors have declared that no competing interests exist.

23.1% and 28% in Indonesia, respectively [4]. Obesity is associated with insulin resistance and complications such as diabetes, hypertension, dyslipidemia, and cardiovascular diseases, which require high medical expenditure and shorten life expectancy [5, 6]. Obese men in particular have shorter life expectancy compared to obese women [6], and a meta-analysis by Wong et al [7] found obesity to be more common in men aged >40 years compared to those below 40.

Obesity is traditionally defined as having a Body Mass Index (BMI) of $\geq$ 25 kg/m$^2$ in Asia Pacific population [8]. However, it was found that BMI is not sufficient to capture the risk of obesity's complications [9]. Waist circumference (WC) is another anthropometric measurement that is a better predictor of metabolic and cardiovascular outcomes due to its close correlation with visceral adipose tissue (VAT) [10]. VAT is a metabolically active site that plays a significant role in the pathophysiology of obesity, and having excessive amount of VAT is defined as visceral or central obesity [11]. A study in Japan found the cut-off point for obesity-related disorders was a VAT area of 100 cm$^2$ [12]. WC cut-off point for central obesity varies across countries, even in Asia. In Indonesia, a country with more than 600 ethnicities [13], there is still no locally developed WC cut-off to diagnose central obesity as Indonesia adopted the WC cut-off points from Japan. This can lead to inaccuracies in predicting a patient's risk for developing obesity-related disorders.

This study aims to find BMI and WC's correlation with VAT in middle-aged Indonesian men, as measured with abdominal CT. In addition, we also developed formulas that predict VAT area using BMI and WC. Next, we also assess the cut-off values for BMI and WC to predict central obesity.

## Research design and methods

This cross-sectional study was conducted at the Radiology Department, Faculty of Medicine Universitas Indonesia/Dr. Cipto Mangunkusumo National General Hospital in Jakarta, Indonesia, which is the central referral hospital in the country. Written informed consent was obtained from all subjects. This study was approved by the Ethical Committee of the Faculty of Medicine Universitas Indonesia (1279/UN2.F1/ETIK/2018).

Sample size was calculated using the general formula for calculating total sample size when using the correlation coefficient [14], with 95% confidence interval (CI) for type I error, and a type II error of 10%. According to a previous study by Camhi et al [15], the correlation coefficient (r) between VAT and WC was 0.73–0.77, whereas the r between VAT and BMI was 0.61–0.69. Using this as a reference, the calculation resulted in a minimum sample size of 25.

Subjects were recruited using time-location sampling as a form of multi-step sampling procedure [16]. All subjects were male patients aged 40–60 years at the Cipto Mangunkusumo National General Hospital who underwent abdominal CT scan from December 2018 to February 2019. The inclusion criteria were patients who were fully alert, cooperative, and able to walk or maintain a standing position. Exclusion criteria were patients with a history of liposuction surgery, massive ascites, significant weight reduction in the past three months, intraabdominal tumor that changed body shape, and paralytic or obstructive ileus. The subjects were scanned while wearing a standard robe for CT scan and had fasted for at least 4 hours prior to the examination.

Primary data was the area of VAT obtained through abdominal CT scan using Siemens Somatom Definition Flash dual-source 128-slice CT scan machine with 120 kV, auto-mAs, with slice thickness of 1.0 mm or Philips Ingenuity 65 64-slice CT machine with 100 kV, auto-mAs, with a slice thickness 1.0 mm; volume measurement software Syngo™ which used Hounsfield Unit (HU) of -30 to -190 HU to select adipose tissue [17–20]; and WC measured

by anthropometric measurement using measuring tape at umbilical level, expressed in centimeter (cm) unit with precision of 1 decimal. BMI was also measured using anthropometric measurement using the subject's height and weight, expressed in kg/m$^2$. Nutritional status was categorized into underweight, normal weight, overweight and obese based on WHO criteria for the Asia Pacific population [8]. VAT and WC measurements were done while the patients were in expiration.

## Statistical analysis

All data were documented in tables and statistically analyzed using SPSS version 20 software. The data underwent Shapiro-Wilk normality test in which Pearson correlation coefficient output (r) was used for normally distributed data, while for abnormally distributed data, we used Spearman correlation coefficient output (ρ) along with its *p* value. In this study, we used conventionally accepted *p* value of 0.05 (confidence level of 95%) as the cut-off point for statistical significance. We performed linear regression analysis to define the formula to predict VAT area based on WC and BMI. To define the cut-off value of central obesity, we performed ROC analysis using VAT area of more than 100 cm$^2$ as the reference [21].

## Results

Thirty-two men aged 40 to 60 years old were recruited in this study, consisting of 14 (43.8%) subjects aged 40–50 years old and 18 (56.2%) subjects aged 50–60 years old (Table 1). There were 2 (6.2%) underweight subjects, 8 (25%) normal weight subjects, and 22 (68.8%) overweight and obese subjects.

There was a significantly strong positive correlation (r = 0.78, p <0.001) between WC and VAT area (Fig 1A). A strong positive correlation (r = 0.67, p <0.001) was also observed between BMI and VAT area (Fig 1B). While the formula to predict VAT area from WC was −182.65 + (3.35 × *WC*), the formula to predict VAT area from BMI was 57.22 + (6.95 × *BMI*). Next, we used ROC curve analysis (Youden index) to determine the cut-off value for central obesity. The AUC for WC and BMI were 0.95 and 0.91, respectively (Fig 2), while the optimal cut-off value for VAT areas of more than 100 cm$^2$ was 88.5 cm and 23.9 kg/m$^2$ for WC and BMI, respectively. From AUC curve, we obtained the sensitivity and specificity for both WC and BMI to determine VAT, which were 81% and 91% for WC, and 81% and 82% for BMI.

**Table 1. Characteristics of subjects.**

| Characteristic | Subjects (n = 32) |
|---|---|
| **Age** (median, interquartile range) | 53 (40–60) |
| **Risk factor** | |
| *Smoking (%)* | 22 (68.8%) |
| *Diabetes melitus (%)* | 7 (21.9%) |
| *Hypertension (%)* | 7 (21.9%) |
| *Dyslipidemia (%)* | 9 (28.1%) |
| **Waist circumference** (cm, mean, SD) | 89.9 (10.3) |
| **Body weight (kg)** | 68.1 (12.8) |
| **Body height (m)** | 1.64 (0.06) |
| **BMI** (kg/m$^2$, mean, SD) | 25.3 (4.5) |
| **Visceral Adipose Tissue** (cm$^2$, median, interquartile range) | 128.9 (31.1–198.6) |

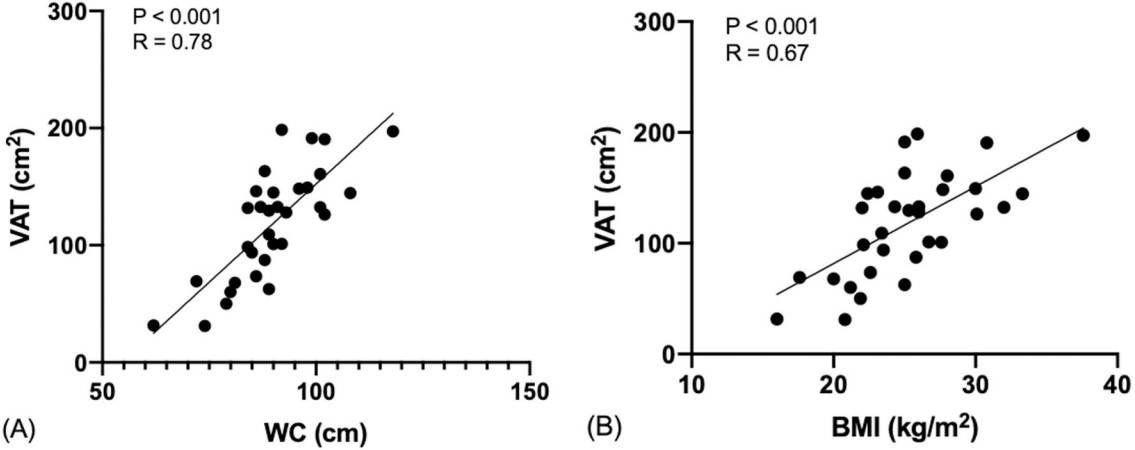

**Fig 1. Scatterplot of the Correlation between VAT and WC (A) and VAT and BMI (B).** The solid lines represent the relationship between two variables obtained from linear regression (statistically significant if p<0.05, with confidence level of 95%).

## Discussion

In this study, we observed strong correlations between WC and VAT area, and also between BMI and VAT area in middle-aged Indonesian men. Next, we developed formulas to predict VAT area using WC and BMI, and found that the cut-off values for central obesity in middle-aged Indonesian men were 88.5 cm and 23.9 kg/m$^2$ for WC and BMI, respectively.

The strong correlations between VAT area and WC (r = 0.78) as well as VAT area and BMI (r = 0.67) confirmed similar findings reported in previous studies [15, 22, 23]. This may suggest that either WC or BMI can be good candidates for being simple anthropometric measures to predict central obesity in our study population, even though the use of BMI alone should be done with caution [24–26]. The simple formulas to predict VAT area using BMI and WC can potentially be applied in low-resource settings where CT scan or MRI are not available.

This study was the first to investigate the cut-off points for central obesity in the middle-aged Indonesian male population-based on VAT area as measured by CT scan. To this day, we

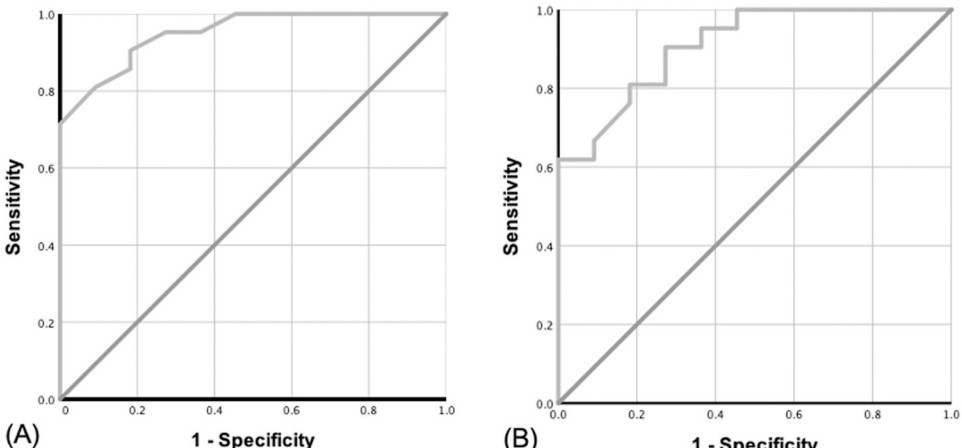

**Fig 2. The predictive value of WC (A) and BMI (B) to predict visceral obesity (VAT ≥ 100 cm$^2$) are presented in ROC.** The AUC of the WC ROC is 0.95 while the AUC of BMI ROC is 0.91.

have not used country-specific cut-off points in Indonesia, and instead used the Asia-Pacific standards. This practice might not provide an accurate diagnosis of obesity. The WC and BMI cut-offs we observed were lower than that of the obesity criteria by WHO Asia-Pacific guideline, which are 90 cm and 25 kg/m$^2$, respectively. Numerous studies done in various countries have shown that Asian populations appear to have higher metabolic risk at lower WC compared to Europeans [27]. In another Asia-Pacific country, Korea, the optimal WC cutoffs for Korean adults were 85 cm for men and 80 cm for women [28]. In China, the WC cut-offs for adults were 80 cm for both men and women [29]. On the other hand, a cross-sectional study Ahmad et al. reported that the optimal cut-off points of WC in Malaysians were somewhere between the Caucasian and Asian cut-off points, which were 92.5 cm for men and 85.5 for women [30].

Our study has several strengths. First, this study analyzed the cutoff for central obesity based on CT scan measurement of VAT area, which is a high-precision method and the gold standard for quantitative assessment of VAT [10]. Previous studies that assessed VAT in Indonesian population used bio-impedance analysis and Dual energy X-ray Absorptiometry (DXA) [31–34], which are not as accurate as CT or MRI [10]. Second, the formulas we developed to predict VAT only require two anthropometric parameters, rendering it a more practical substitute to the calculation by imaging methods which may not be available in rural areas of Indonesia. The use of these formulas, especially in middle-aged men, can assist in preventing the cardiometabolic risks associated with central obesity.

One of the limitations is that our study was limited to men aged 40–60. Further studies on different age and sex groups are needed. Moreover, this study did not investigate the correlation between VAT area and other anthropometric parameters such as waist-hip ratio. It is also important to note that the percentage of people with obese or overweight in this study is higher than the actual situation in Indonesia. Therefore, this study's results might not be generalizable.

## Conclusions

This study found that the WC and BMI cut-offs for central obesity in middle-aged Indonesian men were lower than that of Asian standards. Two formulas have been developed to calculate VAT area using WC and BMI. We propose that studies on different age and sex groups be conducted among the Indonesian population.

## Supporting information

**S1 Data.**
(XLS)

## Author Contributions

**Conceptualization:** Sahat Basana Romanti Ezer Matondang, Bennadi Adiandrian, Dicky Levenus Tahapary.

**Data curation:** Sahat Basana Romanti Ezer Matondang, Dicky Levenus Tahapary.

**Methodology:** Sahat Basana Romanti Ezer Matondang, Bennadi Adiandrian, Komang Shary Karismaputri, Cicilia Marcella, Joedo Prihartono, Dicky Levenus Tahapary.

**Supervision:** Sahat Basana Romanti Ezer Matondang, Dicky Levenus Tahapary.

**Writing – original draft:** Sahat Basana Romanti Ezer Matondang, Bennadi Adiandrian, Komang Shary Karismaputri, Cicilia Marcella, Joedo Prihartono, Dicky Levenus Tahapary.

**Writing – review & editing:** Sahat Basana Romanti Ezer Matondang, Bennadi Adiandrian, Komang Shary Karismaputri, Cicilia Marcella, Joedo Prihartono, Dicky Levenus Tahapary.

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
