## [Decision Letter · Decision Letter 0]

12 Sep 2022

PONE-D-22-18545Simple Anthropometric Measures to Predict Visceral Adipose Tissue Area in Middle-Aged Indonesian MenPLOS ONE

Dear Dr. Tahapary,

Thank you for submitting your manuscript to PLOS ONE. After careful consideration, we feel that it has merit but does not fully meet PLOS ONE’s publication criteria as it currently stands. Therefore, we invite you to submit a revised version of the manuscript that addresses the points raised during the review process.

We look forward to receiving your revised manuscript.

Kind regards,

Yosuke Yamada

Academic Editor

PLOS ONE

Journal Requirements:

“This study is funded by Universitas Indonesia 2Q2 Grant No. NKB-3385/UN2.RST/HKP.05.00/2020.”

“The study is funded by Universitas Indonesia 2Q2 Grant No. NKB-3385/UN2.RST/HKP.05.00/2020.”

“This study is funded by Universitas Indonesia 2Q2 Grant No. NKB-3385/UN2.RST/HKP.05.00/2020.”

Reviewers' comments:

Reviewer's Responses to Questions

**Comments to the Author**

1. Is the manuscript technically sound, and do the data support the conclusions?

Reviewer #1: Yes

Reviewer #2: Partly

2. Has the statistical analysis been performed appropriately and rigorously? 

Reviewer #1: Yes

Reviewer #2: No

3. Have the authors made all data underlying the findings in their manuscript fully available?

Reviewer #1: Yes

Reviewer #2: Yes

4. Is the manuscript presented in an intelligible fashion and written in standard English?

Reviewer #1: No

Reviewer #2: Yes

5. Review Comments to the Author

Reviewer #1: 1. Please use updated reference in line 75. This data is from 2016.

2. Please check if the reference is correct in line 98.

3. In line 98, the author said Indonesia is a country with more than 600 ethnicities, and then in line 117, all the volunteers were recruited from one university. What’s the ethnic diversity of this group? Will it be a good representative group for all Indonesian population?

4. Please insert the sample size in line 116 and what analysis method did you use for this sample size, Power analysis?

5. In line 137 normoweight? or normal weight?

6. Please insert the reference paper in line 151.

7. Why did you choose 40-60 as the age limit of the volunteers? Why not age 20 plus? line 157 nornoweight?

8. It would be good to put in height and weight in table 1. So we know the body weight range for the people. The reason is because one limitation of conducting CT scan is its inapplicability for extremely obese people. 68.8% of the subjects in this study were overweight or obese.

9, In figure 1, please note what is the significant cut point of P value in legend.

10. Figure 2 didn’t mark A,B in the plot.

11. In line 183-193, this paper suggested using WC or BMI for central obesity, how about fat free mass? Have you considered adding FFM as an factor that influences WC or BMI? I suggest that you should adjust your WC or BMI by FFM and then do the regression with VAT again.

12. In line 196, same as comment 3, is this population sample a good representative for all of Indonesia?

Reviewer #2: The purpose of this study is to develop a formula to predict visceral adipose tissue (VAT) area from waist circumference (WC) and BMI and to determine its cutoff value. There have been very few papers on the evaluation of visceral fatty obesity using CT or MRI in Indonesia, and the method of estimating it from abdominal circumference and BMI is a simple and very interesting study. On the other hand, the very small number of subjects (small sample size and male only) is clearly a weak point of this study. In addition, there are many papers examining the relationship between visceral fat and body composition using CT or MRI, but I could not understand why a simple method that does not correct for age, weight, etc. needs to be developed. In addition, considering the large number of previous international studies, this study should have a much larger number of cited papers.

Major comments

The importance of studies participated in men should be added to Line73-86. Similarly, the importance of assessing visceral fat should be added to Lines 91-94.

It should be emphasized that most of the previous studies in Indonesia used simple methods such as the impedance method, and few studies have used in CT or MRI of high-precision method. The following paper assessing visceral fat in Indonesia may be helpful to cite.

doi: 10.1016/j.orcp.2020.11.003.

doi: 10.3390/jcm7050096

doi: 10.1016/j.diabres.2007.01.062.

doi: 10.2337/dc18-1074.

Line 116 Were the subjects recruited from patients attending the hospital? Please add the method of recruiting subjects.

Line 125 and Line 134. What conditions were body composition and CT assessments measured? Please describe the measurement conditions, such as clothing, fasting time, etc.

Line 131. Please add a citation to show the rationale for setting the HU values for fat in the range of -190 to -30 HU.

6. PLOS authors have the option to publish the peer review history of their article (what does this mean?). If published, this will include your full peer review and any attached files.

Reviewer #1: No

Reviewer #2: No

---

## [Author Response · Author response to Decision Letter 0]

25 Oct 2022

Thank you afor all the valuable comments and suggestions provided by the editors and reviewers. We have considered all the comments and suggestions carefully and have implemented the suggestions as much as possible. 

The manuscript has been rechecked, and the necessary changes have been made in accordance with the reviewers' suggestions. The responses to all comments have been prepared and are attached as a document titled "Response to Reviewers".

A few points that we would like to make:

- The subjects of this study were recruited using time-location sampling as a form of multi-step sampling procedure (Leon et al, 2015). The time of sampling was December 2018 to February 2019, whereas the location was the Cipto Mangunkusumo National General Hospital. The location is representative of Indonesian population because it is a national referral hospital. The method of time-location sampling can be non-biased if we considered frequency of venue attendance (Leon et al, 2015). In this study, we considered FVA by including all respondents who were willing to participate during the time of sampling at the Cipto Mangunkusumo National General Hospital.

- For the sample size, we used the general formula for calculating total sample size when using the correlation coefficient and acquired a minimum sample of 25. 

- We conducted this research only on males because the risk of fatality caused by obesity is more significant in males compared to females. This can be observed by the life expectancy of obese males which is significantly less than that of obese females (Nagai et al, 2015). 

- Regarding the age of our samples, we decided to include only males aged 40-60. According to a meta-analysis by Wong et al (2020), the prevalence of central obesity for men is higher in subjects aged > 40 years. This research did not include men aged >60 years because in Indonesia they are categorized as geriatric patients. Moreover, according to research by Schousboe et al (2018), central obesity and VAT are not associated with atherosclerotic cardiovascular disease events in men aged more than >65 years. 

Below are the additional references: 

1. Hulley SB. Designing Clinical Research. 5th edition. Philadelphia: Lippincott Williams & Wilkins. 367 p. 

2. Camhi SM, Bray GA, Bouchard C, Greenway FL, Johnson WD, Newton RL, et al. The Relationship of Waist Circumference and BMI to Visceral, Subcutaneous, and Total Body Fat: Sex and Race Differences. Obesity (Silver Spring). 2011 Feb;19: 402-8.

3. Nagai M, Kuriyama S, Kakizaki M, Ohmori-Matsuda K, Sone T, Hozawa A, et al. Impact of obesity, overweight and underweight on life expectancy and lifetime medical expenditures: the Ohsaki Cohort Study. BMJ Open. 2012;2:e000940.

4. Wong MCS, Huang J, Wang J, Chan PSF, Lok V, Chen X, et al. Global, regional and time‑trend prevalence of central obesity: a systematic review and meta‑analysis of 13.2 million subjects. Eur J Epidemiol. 2020; 35:673-83. 

5. Republic of Indonesia. Law of the Republic of Indonesia number 13 in 1998 concerning elderly welfare. Jakarta (ID): Republic of Indonesia; 1998. 24 p. 

6. Schousboe JT, Kats AM, Langsetmo L, Vo TN, Taylor BC, Schwartz AV. Central Obesity and Visceral Adipose Tissue Are Not Associated With Incident Atherosclerotic Cardiovascular Disease Events in Older Men. J Am Heart Assoc. 2018;7:e009172.

7. Leon L, Jauffret-Rouside M, Strat YL. Design Design-based inference in time-location sampling. Biostatistics. 2015;16:565-79.

---

## [Decision Letter · Decision Letter 1]

29 Nov 2022

PONE-D-22-18545R1Simple Anthropometric Measures to Predict Visceral Adipose Tissue Area in Middle-Aged Indonesian MenPLOS ONE

Dear Dr. Tahapary,

Thank you for submitting your manuscript to PLOS ONE. After careful consideration, we feel that it has merit but does not fully meet PLOS ONE’s publication criteria as it currently stands. Therefore, we invite you to submit a revised version of the manuscript that addresses the points raised during the review process.

We look forward to receiving your revised manuscript.

Kind regards,

Yosuke Yamada

Academic Editor

PLOS ONE

Journal Requirements:

Reviewers' comments:

Reviewer's Responses to Questions

**Comments to the Author**

1. If the authors have adequately addressed your comments raised in a previous round of review and you feel that this manuscript is now acceptable for publication, you may indicate that here to bypass the “Comments to the Author” section, enter your conflict of interest statement in the “Confidential to Editor” section, and submit your "Accept" recommendation.

Reviewer #1: All comments have been addressed

Reviewer #2: All comments have been addressed

2. Is the manuscript technically sound, and do the data support the conclusions?

Reviewer #1: Yes

Reviewer #2: Yes

3. Has the statistical analysis been performed appropriately and rigorously? 

Reviewer #1: Yes

Reviewer #2: Yes

4. Have the authors made all data underlying the findings in their manuscript fully available?

Reviewer #1: Yes

Reviewer #2: Yes

5. Is the manuscript presented in an intelligible fashion and written in standard English?

Reviewer #1: Yes

Reviewer #2: Yes

6. Review Comments to the Author

Reviewer #1: The authors have adequately addressed my comments raised in the previous round of review, I have no other comments.

Reviewer #2: Thank you for your sincere response to my comments. I confirmed that this manuscript has been a significant improvement. No additional comments from me if the following areas are improved.

Minor comments

Line 29 It should be changed to "The diagnosing of central obesity...".

Line 131 It should be changed from “healthy weight” to “normal weight”

The percentage of overweight or obese persons in this study (68.8%) appears to be higher than the actual situation in Indonesia. Therefore, the results obtained in this study may not be generalizable. This point should be added to the limitations.

7. PLOS authors have the option to publish the peer review history of their article (what does this mean?). If published, this will include your full peer review and any attached files.

Reviewer #1: No

Reviewer #2: No

---

## [Author Response · Author response to Decision Letter 1]

12 Dec 2022

Comment 1: Line 29 It should be changed to "The diagnosing of central obesity...".

Thank you for the correction. We have revised Line 29 to “The diagnosing of central obesity...” 

Comment 2: Line 131 It should be changed from “healthy weight” to “normal weight”

Thank you for the correction. We have changed the term “healthy weight” to “normal weight” in Line 132 and 154. 

Comment 3: The percentage of overweight or obese persons in this study (68.8%) appears to be higher than the actual situation in Indonesia. Therefore, the results obtained in this study may not be generalizable. This point should be added to the limitations.

Thank you for the suggestion. We have added this limitation in Line 236-239:

“It is also important to note that the percentage of people with obese or overweight in this study is higher than the actual situation in Indonesia. Therefore, this study’s results might not be generalizable.”

---

## [Editor Report · Decision Letter 2]

21 Dec 2022

Simple Anthropometric Measures to Predict Visceral Adipose Tissue Area in Middle-Aged Indonesian Men

PONE-D-22-18545R2

Dear Dr. Tahapary,

We’re pleased to inform you that your manuscript has been judged scientifically suitable for publication and will be formally accepted for publication once it meets all outstanding technical requirements.

Kind regards,

Yosuke Yamada

Academic Editor

PLOS ONE
---

## [Editor Report · Acceptance letter]

27 Dec 2022

PONE-D-22-18545R2 

Simple anthropometric measures to predict visceral adipose tissue area in middle-aged Indonesian men 

Dear Dr. Tahapary:

I'm pleased to inform you that your manuscript has been deemed suitable for publication in PLOS ONE. Congratulations! Your manuscript is now with our production department. 

Kind regards, 

on behalf of

Dr. Yosuke Yamada 

Academic Editor

PLOS ONE